# Molecular basis of differential 3′ splice site sensitivity to anti-tumor drugs targeting U2 snRNP

Luisa Vigevani[1,2], André Gohr[1], Thomas Webb[3], Manuel Irimia[1,2] & Juan Valcárcel[1,2,4]

Several splicing-modulating compounds, including Sudemycins and Spliceostatin A, display anti-tumor properties. Combining transcriptome, bioinformatic and mutagenesis analyses, we delineate sequence determinants of the differential sensitivity of 3′ splice sites to these drugs. Sequences 5′ from the branch point (BP) region strongly influence drug sensitivity, with additional functional BPs reducing, and BP-like sequences allowing, drug responses. Drug-induced retained introns are typically shorter, displaying higher GC content and weaker polypyrimidine-tracts and BPs. Drug-induced exon skipping preferentially affects shorter alternatively spliced regions with weaker BPs. Remarkably, structurally similar drugs display both common and differential effects on splicing regulation, SSA generally displaying stronger effects on intron retention, and Sudemycins more acute effects on exon skipping. Collectively, our results illustrate how splicing modulation is exquisitely sensitive to the sequence context of 3′ splice sites and to small structural differences between drugs.

[1] Centre for Genomic Regulation (CRG), The Barcelona Institute of Science and Technology, Dr Aiguader 88, 08003 Barcelona, Spain. [2] Universitat Pompeu Fabra, Dr Aiguader 88, 08003 Barcelona, Spain. [3] Division of Biosciences, SRI International, 333 Ravenswood Avenue, Menlo Park 94025 CA, USA. [4] Institució Catalana de Recerca i Estudis Avançats (ICREA), Pg. Lluís Companys 23, 08010 Barcelona, Spain. Correspondence and requests for materials should be addressed to J.Vár. (email: juan.valcarcel@crg.eu)

Intron removal from pre-mRNAs is achieved by the spliceosome, composed of five snRNPs (small nuclear ribonucleoproteins: U1, U2, U4, U5 and U6 snRNPs) and >150 polypeptides. Spliceosomes assemble on intron boundaries by recognizing specific sequence signals, the 5′ and 3′ splice sites (ss). The 3′ss encompasses a branch point sequence (or BP), a stretch of pyrimidines (polypyrimidine(Py)-tract) and an AG

dinucleotide at the 3′ end of each intron[1]. Splicing catalysis occurs in two steps. First, the nucleotide at the 5′ end of the intron is covalently bound through a 2′–5′ phosphodiester bond to an adenosine located within the BP sequence, generating a free 5′ exon and a lariat intron. Second, the free 5′ exon is ligated to the 3′ exon and the lariat is excised. Regulation of ss recognition generates alternative patterns of intron removal (alternative

splicing, AS) and distinct messenger RNA (mRNAs) and proteins from a single primary transcript[1–3].

5′ss sequences are initially recognized by U1 snRNP, while the proteins SF1, U2AF65 (or U2AF2) and U2AF35 (or U2AF1) recognize the BP sequence, the Py-tract, and the AG dinucleotide, respectively. Cooperative binding of these proteins helps to recruit U2 snRNP to the 3′ss[1,3], which involves base pairing interactions between its RNA component (U2 snRNA) and the BP region[1] as well as sequence-independent interactions between the region 5′ of the BP ("anchoring site") and components of the U2 snRNP multiprotein subcomplexes SF3A and SF3B[4]. The largest SF3B subunit, SF3B1, encompasses an unstructured N-terminal domain involved in protein–protein interactions and 20 carboxy-terminal HEAT (huntingtin, elongation factor 3, protein phosphatase 2A, target of rapamycin 1) repeats[5]. SF3B1 undergoes phosphorylation in several residues of the N-terminal domain right before the first catalytic step[6]. Recent structural insights from yeast spliceosomes showed that SF3B1's homolog Hsh155 and its HEAT repeats play key roles in presenting the BP adenosine for catalysis[7,8].

Several disease-associated mutations reside within regulatory sequences important for splicing, or within genes encoding splicing factors[9–12]. Mutations in the splicing factors U2AF35, ZRSR2, SRSF2, and SF3B1 are associated with myelodysplastic disorders, chronic lymphocytic leukemia, and solid tumors, including uveal melanoma, with SF3B1 being the most frequently mutated spliceosome component[9,11]. Mutated SF3B1, SRSF2, and U2AF1 cause sequence-dependent AS changes[13–16]. In particular, SF3B1 mutations within the HEAT repeats involving changes in electric charge[5] induce the activation of cryptic 3′ss due to a shift in BP usage[13,14] likely due to altered electrostatic interactions with the RNA.

Several families of compounds with anti-tumor properties (e.g., FR901463-5, GEX1, and the pladienolides), their derivatives (e. g., spliceostatin A or SSA) and their totally synthetic analogs (e.g., meayamycin and sudemycins) target the SF3B complex[9]. The drug SSA, which directly binds SF3B1[17], induces U2 snRNA base pairing with "decoy" sequences 5′ of the conventional BP[18]. The drug E7107 (a derivative of pladienolide) alters the balance between alternative U2 snRNA conformations[19]. In addition, specific residues in SF3B1 and PHF5A (also known as SF3B7 and SF3b14b) embrace the BP adenosine and their mutation leads to resistance to pladienolide and related drugs[20].

Interestingly, sensitivity to SF3B1 inhibitors is increased in cells bearing mutations in SF3B1[21], SRSF2[22], U2AF1[23] and in cells overexpressing cMYC oncogene[24,25], suggesting that interfering with the spliceosome machinery can be particularly effective under conditions where splicing becomes rate-limiting, thus offering potential therapeutic opportunities for cancer[11]. While clinical trials for E7107 (a pladienolide variant) were interrupted

due to ocular toxicity in a subset of patients[11], clinical studies with H3B-8800 are under way to specifically target cancer cells with mutations in SF3B1[26].

One major question is why and how drugs targeting SF3B components, which would in principle affect essential steps in 3′ss recognition and therefore affect every cellular function, can selectively inhibit the growth of cancer cells. In this work, we dissect sequence elements (and their combinations) that modulate differential drug sensitivity, with even subtle sequence differences leading to differential effects on AS that affect cell proliferation and viability. Furthermore, we document differential effects of structurally similar drugs, revealing surprising plasticity in the response to splicing modulatory compounds.

## Results

**Sequence elements 5′ of the BP modulate drug inhibition.** To investigate the sequence determinants of the differential sensitivity of particular introns to SF3B1-targeting drugs, we selected two AS events that differentially respond to Sudemycin C1 (Sud C1). Drug treatment strongly induced skipping of exon 2 of endogenous *MCL1* transcripts (Fig. 1a), consistent with previous results[20,27–29] and with a key role for this gene in drug-mediated induction of apoptosis[27,28] (Supplementary Fig. 1). In contrast, under the same conditions, skipping of *PDCD10* exon 7 was only weakly affected by the drug[18] (Fig. 1a).

These differential effects were recapitulated using minigenes corresponding to the alternatively spliced regions of *MCL1* and *PDCD10* (Fig. 1a, right), arguing that differential drug sensitivity depends exclusively on their primary sequences. Adenosine −25 was determined as the BP of *PDCD10* intron 6 (Supplementary Fig. 2) and inspection of the sequences 5′ of the BP identified three sequence elements (E1-E3, Fig. 1b, left) characterized by the presence of potential BP "YUNAY"[30] motifs followed by a pyrimidine-rich sequence and, in E1 and E3, a AAATGT motif. Consecutive deletion of these elements resulted in progressive increases in drug sensitivity (Fig. 1b, right), indicating that sequences 5′ of the BP can strongly influence drug responses and display additive effects.

To test whether the drug resistance-inducing effects of these elements could be transferred to other transcripts, E3-E2-E1 sequences were inserted 5′ of the BP of *MCL1* intron 1, replacing a sequence of similar length located at an equivalent position in the pre-mRNA. Deletion of the replaced sequence did not have a significant effect on drug response (Fig. 1c, compare MCL1 wt and MCL1 Δ minigenes), nor did its replacement by a sequence of PDCD10 5′ of the E3-E2-E1 elements (Fig. 1c, MCL1 Δ + up PDCD10). In contrast, insertion of the E3-E2-E1 element resulted in strong reduction of the effects of Sud C1 on *MCL1* exon 2 skipping (Fig. 1c, MCL1 Δ + E3-E2-E1), confirming that these

**Fig. 1** Mapping of sequence elements within *PDCD10* intron 6 that confer resistance to Sud C1. **a** Differential effects of Sud C1 on the regulation of *MCL1* exon 3 and *PDCD10* exon 7 skipping. RNAs isolated from HeLa cells exposed to DMSO (as control) or 20 μM Sud C1 for 8 h were analyzed by RT-PCR followed by capillary electrophoresis. Percent spliced in (PSI) average and s.d. values for the quantification of triplicated experiments are shown. Analyses were carried out for endogenous transcripts as well as for transcripts derived from minigenes expressing the corresponding alternatively spliced regions after 16 h of transfection (the latter using primers complementary to transcribed vector sequences). **b** *PDCD10* intron 6 sequence elements 5′ of the BP confer resistance to Sud C1. Three related sequence elements (E1-E3) were distinguished (left), each containing a potential branch point (BP) consensus (bold) and a pyrimidine (Y)-rich element, 2 of them containing also a AAATGT motif. Minigenes containing wild-type (wt) or progressive deletion of the E1-E2-E3 elements were transfected and analyzed as in **a**. **c** Sequence elements associated with Sud C1 resistance in *PDCD10* RNAs confer resistance to *MCL1* transcripts. *MCL1* wild-type and mutant minigenes schematized on the left were assayed as in **a** and the results shown on the right panel. In this case, statistical significance refers to the results of two-sided *t* test comparisons between PSI values of Sud C1-treated cells expressing mutated vs wt minigenes. *; **; ***: two-sided *t* test *P* value < 0.01, 0.001 and 0.0001, respectively. n.s.: non-significant (*P* > 0.01). A minimum of three treatment replicates was used for each condition. Error bars represent standard deviation from the mean value

sequence elements are sufficient to inhibit the effects of the drug on AS.

To further dissect the contribution of sequence elements within "E" regions, we separately tested the potential effects of the BP consensus and pyrimidine-rich sequences. Remarkably, insertion of a sequence containing a BP consensus (CTCTCAC) led to inhibition of the effects of Sud C1 almost as strong as the insertion of the E3-E2-E1 elements (Fig. 1c, MCL1 Δ + BP). Importantly, deletion of the adenosine in this sequence element, which is critical for BP function in 2′–5′ phosphodiester bond formation, completely abrogated this effect, indicating that a decoy BP (i.e., a sequence able to base-pair with U2 snRNA but inactive for splicing) fails to inhibit drug response (Fig. 1c, MCL1 Δ + decoy).

Interestingly, insertion of the sequence TTTCTTTCAAATGT from the E1 element also reduced drug effects (Fig. 1c, MCL1 Δ + Y-rich sequence + AAATGT). The combination of this element and a BP consensus (even in a non-natural order) resulted in full inhibition of the effects of the drug (Fig. 1c, MCL1 Δ + Y-rich + AAATGT + BP). While the pyrimidine-rich sequence by itself did not affect drug response significantly (Fig. 1c, MCL1 Δ + Y-rich), the AAATGT element (which harbors additional potential sites for BP formation) did (Fig. 1c, MCL1 Δ + AAATGT).

Collectively, the results argue that the presence of additional functional BP sequences 5′ of the natural BPs of MCL1 intron 2 attenuate the splicing effects of drugs targeting SF3B1.

**Human MCL1 intron 1 sensitivity to SF3B1-targeting drugs.** To obtain further insights into sequence features responsible for the special sensitivity of MCL1 exon 2 to Sud C1, we determined sequences in MCL1 intron 1 that serve as BP. First, we isolated PCR amplification products specific for lariat RNAs that join together the 5′ss and the BP region across the 2′-5′ phosphodiester bond[30]. Several potential BPs in the region −19 to −33 of MCL1 intron 1 were identified in RNAs from HeLa cells (Fig. 2a and Supplementary Fig. 2), possibly including non-adenosine residues, as previously observed in transcriptome-wide BP mapping approaches[30,31].

A relevant insight came from the observation that MCL1 exon 2 is not alternatively spliced in mouse cells. Although mouse MCL1 transcripts use alternative ss within exon 1[32], to date, no evidence of transcripts skipping exon 2 has been annotated. To explore this further, patterns of MCL1 splicing in the absence or presence of Sud C1 were investigated. No effect was observed in endogenous mouse Mcl1 transcripts (Supplementary Fig. 2e) or using minigenes carrying the relevant mouse genomic sequences, transfected in human HeLa cells or in mouse 3T3 cells (Fig. 2b). The results indicate that mouse Mcl1 transcripts are refractory to the exon skipping effects of Sud C1, while human MCL1 sequences are highly responsive. These differences are observed both in human and in murine cells (Fig. 2b), indicating that the differential effects of the drug depend on the primary sequences of these transcripts rather than on differences between the splicing machinery of mouse and human cells.

Next, we generated chimeric minigenes where sequences of the 3′ region of mouse intron 1 were replaced by the corresponding sequences of the human intron. Replacements included the region 5′ of the putative BP (from −104 to −29 bp from the 3′ss, the predicted BP adenosine being located at position −23) or the entire 3′ss region (from −104 to the 3′ end of the intron, including the putative BP and Py-tract). While replacement of the region 5′ of the BP did not confer drug sensitivity to the mouse minigene-derived transcripts, replacement of the 3′ss region did (chimeras 1 and 2, Fig. 2b). These results suggest that classical sequence

features of the 3′ss contribute to the differential effects between the mouse and human transcripts. Therefore, we carried out mutational analysis of the predicted hMCL1 BP sequences. Adenosine at position −26 of MCL1 intron 1 was identified as one of the possible BPs[31], but mutation to cytosine showed only a small effect on exon inclusion and the mutation did not affect drug sensitivity (Fig. 2c, hMCL1 A-26C). In contrast, deletion of the ACCTGCA sequence element located between positions −23 and −17 significantly reduced the levels of exon inclusion even in control conditions (Fig. 2c, hMCL1 ΔACCTGCA), arguing that these sequences play an important role in efficient splicing of human MCL1 intron 1, possibly acting as the main BPs for the formation of lariats during intron 1 excision (Supplementary Fig. 2c). Sud C1 treatment resulted in further reduction of exon 2 inclusion, arguing that the drug efficiently inhibits both the mutated 3′ss (using alternative BPs) as well as the cryptic 3′ss activated by the mutation (Supplementary Fig. 2d). In contrast, replacement of the nucleotides around adenosine −26 by a consensus BP sequence TACTAAC, which works as the optimal BP functional sequence from yeast to mammals[33], resulted in efficient exon 2 inclusion and, importantly, complete resistance to the effects of the drug (Fig. 2c, hMCL1 −31/-25 TACTAAC). These results argue that the strength of the BP sequence is a determinant of drug response and, consequently, that the presence of multiple rather weak BPs sequences in human MCL1 intron 1 confers sensitivity to Sud C1 treatment.

Three noticeable differences between human and mouse 3′ss regions were observed (Fig. 2a): (1) a longer Py-tract in the mouse intron, (2) absence in the mouse gene of the ACCTGCA sequence, which includes several potential BP sequences and (3) a single nucleotide difference at position −29 of the human sequence, where a guanosine is replaced by adenosine in the mouse gene. Mutation of G −29 to A in the human minigene resulted in some decrease in the effect of Sud C1 treatment, but the converse mutation (A−G) at the equivalent position of the mouse 3′ss region did not affect the drug resistance of the mouse minigene (Fig. 2c, hMCL1 G-29A and mMCL1 A-26G). Replacement of the Py-tract of the human sequence by the longer Py-tract of the mouse sequence (underlined nucleotides in Fig. 2a) abolished the effects of the drug (Fig. 2c, hMCL1 mPY). This result was maintained after deletion of the −23/−17 ACCTGCA sequence (Fig. 2c, mMCL1 mPY ΔACCTGCA), suggesting that the presence of the longer Py-tract suppresses the effects of the drugs in the absence of one of the clusters of BPs found important for the function of the human 3′ss (Supplementary Fig. 2), further arguing for a key role of Py-tract length and/or its specific sequence in conferring drug resistance.

Collectively, our results indicate that the strength/degeneracy of BPs, as well as the length/strength of the Py-tract are determinants of the sensitivity of 3′ss to treatment with SF3B1-targeting drugs. Furthermore, the presence of decoy or bona fide BP mimics, located 5′ of the functional BP, can also modulate drug response.

**Sequence features of Sud C1-induced exon skipping events.** To further analyze sequence features influencing Sud C1-induced regulation at transcriptome-wide level, we performed RNA-Seq analysis. Regulation of splicing isoforms is complex and steady-state levels can be influenced by transcript stability and other indirect effects, including mRNA transport, as reported recently for SSA-treated cells[34]. To analyze recently transcribed transcripts (which in principle provide a more immediate view of nascent/recent splicing events), cultured HeLa cells were co-treated with 10 μM Sud C1 and 2 mM 5-Bromouridine (BrU) for 3 h (a

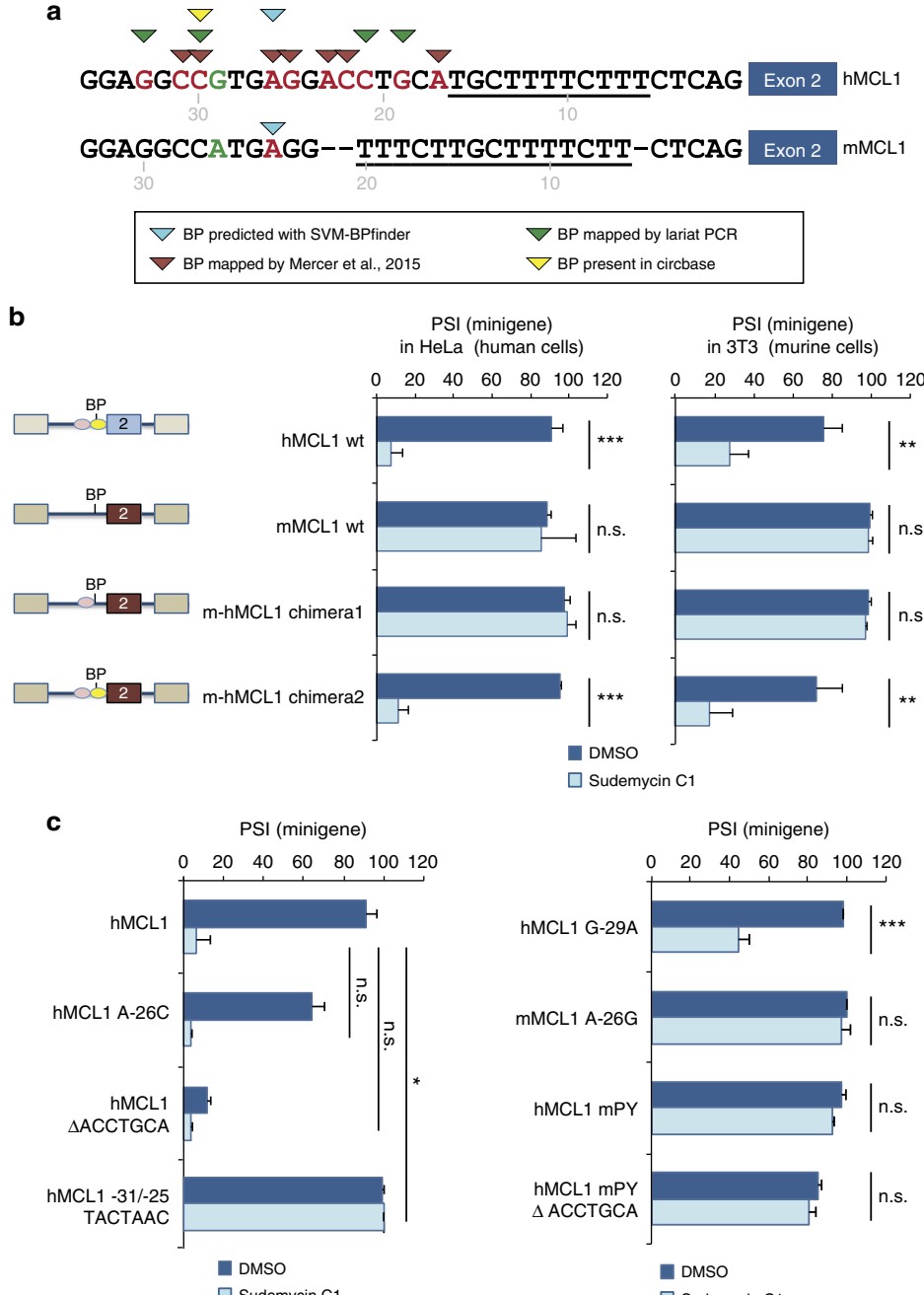

**Fig. 2** Differential regulation of human and mouse *MCL1* reveals a role for Py-tracts in modulating drug response. **a** Sequences of the 3′ end of human (h) and mouse (m) *MCL1* intron 1. Underlined residues indicate the Py-tracts. G/A in green indicates a sequence difference in the 5′ part of the region. Residues in red correspond to the predicted BP (for mouse *MCL1*) or mapped BPs (for human *MLC1*) using various strategies, as indicated by the triangles' color code. **b** Distinct effects of Sud C1 on minigenes expression of human or mouse *MCL1* transcripts. Minigenes corresponding to the human, mouse or chimeric constructs (containing the indicated regions of human intron 1 replacing the corresponding regions of the mouse transcripts) were analyzed in HeLa (human) or 3T3 (mouse) cell lines as in Fig. 1. Chimera 1: positions −104 to −29 from the 3′ss AG of the mouse gene were replaced by sequences −103 to −28 of the human gene. Chimera 2: 103 nt of the 3′ end of the mouse intron were replaced with the corresponding sequences of the human gene, which include the human BP and Py-tract. Sequences are indicated in Supplementary Table 1. **c** Assessment of the contributions of various sequence elements to Sud C1 response: Assays were carried out as in **b** for the indicated mutant constructs. *; **; ***: two-sided *t* test *P* value < 0.01, 0.001 and 0.0001, respectively. n.s.: not statistically significant (*P* > 0.01). A minimum of three treatment replicates was used for each condition. Error bars represent s.d. from the mean value

time frame that allows detection of recently transcribed RNAs, both unspliced and spliced), and BrU-containing RNAs isolated by immunoprecipitation with a specific antibody[35]. BrU was found not to induce significant changes in the splicing patterns of fully processed RNAs.

RNA-seq analysis of total RNA and recently transcribed RNA revealed widespread exon skipping and intron retention (Fig. 3a), with over 1400 exon skipping events and 1600 intron retention events in total RNA and over 3700/9600 events in recently transcribed RNA. The significantly higher number of regulated

transcripts (particularly intron retention) in BrU-labeled RNA may reflect more extensive splicing alterations in recently transcribed RNAs, with a substantial number of RNAs displaying splicing changes being subsequently eliminated by quality control mechanisms. However the vast majority of drug-induced events detected in total RNA were also detected in BrU-labeled RNA, arguing that BrU labeling does not affect substantially the effects of the drug. Gene ontology (GO) analysis revealed that genes displaying splicing changes are mainly involved in key cellular functions such transcription, RNA processing, cell metabolism, and cell cycle control (Supplementary Figs. 3a, b).

Mechanistic connections between transcription and splicing are strong and recent studies have reported that elevated cMYC levels increase the sensitivity of cancer cells to splicing inhibition[24,25], suggesting the possibility that high levels of transcription require high levels of splicing; this connection could be one of the main reasons for the higher sensitivity of tumors to SF3B1 inhibition. We therefore analyzed whether there was a

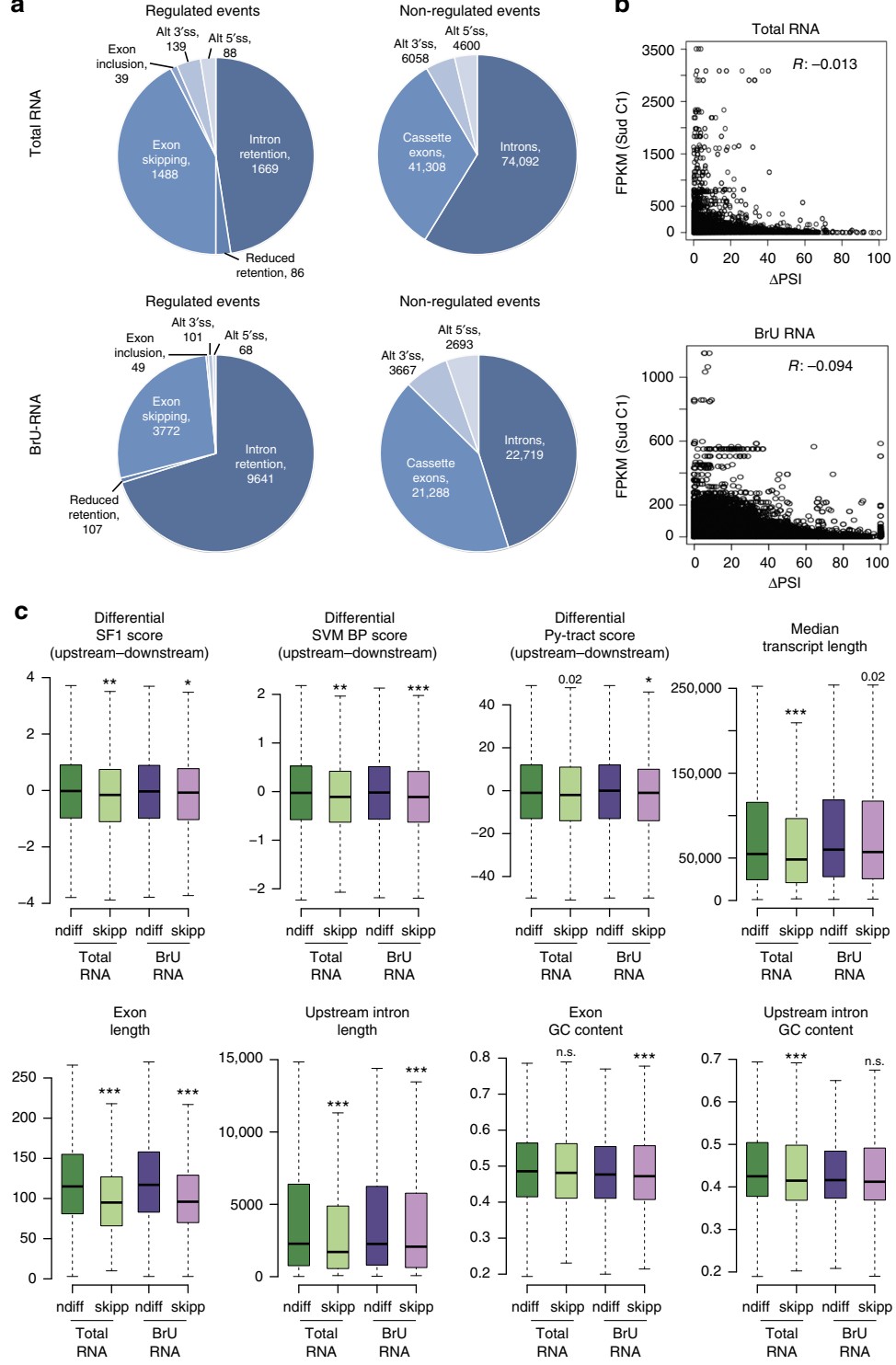

correlation between the levels of gene expression and the effects of Sud C1 on specific splicing events (both intron retention and exon skipping) within a gene. No significant correlation was found for either total or recently transcribed BrU-labeled RNA (Fig. 3b), further supporting that transcript-specific sequence features are the main determinants of drug sensitivity.

The analysis of sequence elements associated with drug-induced exon skipping revealed that: (i) 3′ss associated with skipped exons display significantly weaker BPs, as determined by SVM-BPfinder and SF1 binding site scores[36,37], and marginally weaker Py-tracts compared to the 3′ss with which they are in competition (located in the corresponding downstream intron) (Fig. 3c, upper panels); (ii) the length of the exon and of the upstream intron are significantly shorter in drug-induced skipped exons than in other alternative exons not affected by the drug treatment, both in total and BrU-labeled RNA samples; in contrast, only a slight tendency toward lower exon GC content and transcript length was observed (Fig. 3c, lower panels).

**Sequence features of Sud C1-induced intron retention events.** Intron retention is the most abundant splicing alteration induced by Sud C1 (Fig. 3a). A comparison of sequence elements of drug-induced retained introns compared to non-affected introns revealed that: (i) drug-retained introns generally display weaker Py-tract and BP signals, both in BrU RNA and in total RNA (Fig. 4, upper panels). These data are reminiscent of the results of minigene mutagenesis for exon skipping events (Figs. 1 and 2); (ii) analysis of experimentally mapped BPs (Taggart et al., [38]) revealed that affected sequences tend to have more distant and unconventional BPs compared to non-affected ones (Supplementary Fig. 4); (iii) consistent with our observation that additional potential BP sequences located 5′ of the BP can confer drug resistance (Figs. 1 and 2), significantly weaker effects on intron retention and cassette exon skipping were observed for 3′ss containing more than one match with the consensus YNYYRAY BP within the 100 3′ nucleotides of the intron (Fig. 4b; RNA maps in Supplementary Fig. 5); (iv) drug-retained introns are significantly shorter and more GC-rich in both BrU-labeled and total RNA (Fig. 4a, lower panels). Of potential relevance for these observations, previous studies reported lower nucleosome occupancy in exons neighboring short and GC-rich introns, which are indeed prone to intron retention. These results correlate with the proposed role of nucleosomes in exon definition[39–41] and with the lower recruitment of SF3B1 to nucleosomes located in exons flanked by short and GC-rich introns[42,43]. Consistently, drug-retained introns occurred in generally shorter transcripts (Fig. 4, right lower panel; examples in Supplementary Fig. 3).

Collectively, the data in Figs. 3 and 4 support an important role of intronic sequence features for Sud C1-induced intron retention, as summarized in Fig. 4c.

**Diverse AS modulation induced by different drug variants.** SSA, Sud C1, and the recently described Sud K[44] have related structures but show some chemical differences (Fig. 5a). The compounds require different concentrations to reach comparable effects, as previously reported[44,45], consistent with significant differences in the concentrations required to affect the viability of cultured cells upon SSA, Sud C1 (Fig. 5b), or Sud K treatment[44].

Previous work showed that SSA destabilizes complex A formation on 3′ss (A3′ complex) in the presence of heparin[18]. To test whether a similar mechanism operates in Sud C1- and Sud K-mediated splicing regulation, a radioactively labeled RNA corresponding to the 3′ end of adenovirus major later (AdML) promoter intron 1 and second exon was incubated in HeLa nuclear extracts in the absence or presence of the drugs. The three drugs significantly inhibit A3′ complex formation in the presence of heparin (Fig. 5c), albeit at different concentrations (Fig. 5c), indicating that the different activities of SSA and Suds are not simply due to differences in cellular permeability, but are rather associated with structural differences that impact on the activity or stability of these compounds.

To further explore the similarities and differences in the activities between these drugs, we compared their effects in both steady-state RNAs as well as recently transcribed RNAs purified after a 3 h pulse with BrU. Drug-induced skipping of MCL1 exon 2 was found to be similar in BrU pulse-labeled and in steady-state RNA (Fig. 5d, left), suggesting that the splicing switch is the main contributor to the change in isoform ratios observed in steady-state transcripts. Similar effects were observed in another AS event in the gene ARRDC3, which we found to be affected by Sud C1 in previous splicing-sensitive microarray analyses. Interestingly, while SSA, Sud C1 ans Sud K induced similar levels of MCL1 exon 2 skipping, SSA induced higher levels of ARRDC3 exon 3 skipping (Fig. 5d). Drug titration in a range of concentrations showed that while SSA generally displayed weaker effects than Sud C1 or Sud K on MCL1 exon 2 skipping across the range of concentrations tested, SSA displayed stronger effects on ARRDC3 exon 3 skipping (Fig. 5e). Collectively, the results shown in Fig. 5d, e indicate that structurally similar drugs can display distinct effects on different AS events.

To expand these analyses, RNA-seq was carried out using both total RNA as well as BrU-labeled RNA. Drug concentrations were adjusted to display similar effects on MCL1 exon 2 skipping on the basis of previous experiments (Fig. 5). As was the case for Sud C1, SSA, and Sud K induced mainly intron retention and exon skipping (Supplementary Fig. 6), with highly significant overlaps in changes detected in total and BrU RNA (Supplementary Fig. 7). On the other hand, the analysis unveiled multiple differences in the effect of the three drugs. First, SSA and Sudemycins showed differential preference for intron retention and exon skipping, respectively. While similar numbers of exon skipping events were induced in BrU RNA by the three drugs

**Fig. 3** Transcriptome-wide analysis of Sud C1-induced splicing regulation. **a** Intron retention and cassette exon skipping are the AS categories most affected by Sud C1. Number and distribution of categories of regulated AS events, with I∆PSII ≥ 25 (∆PSI: PSI of treated sample–PSI of control sample, where PSI indicates the percent spliced in) and I∆PIRI ≥ 25 (∆PIR: PIR of treated sample–PIR of control sample, where PIR indicates the percent of intron retention). Non-regulated events: I∆PSII and I∆PIRI ≤ 5. Splicing was quantified using VAST-TOOLS (Methods) by analyzing RNA-Seq data of total and BrU RNAs from cells treated for 3 h with 10 µM Sud C1 vs DMSO. **b** Lack of correlation between gene expression and AS changes. Plots represent I∆PSII or I∆PIRI (AS changes) values vs FPKM (transcription) values derived from VAST-TOOLS analysis of RNA-Seq data from total or BrU-pulsed RNAs. R indicates correlation coefficient. **c** Sequence features with significant changes between Sud C1-induced cassette exon skipping and non-differentially spliced exons. Boxplots represent the distribution of feature values (Methods) in non-differentially spliced (ndiff) (I∆PSII ≤ 5) and skipped (skipp) cassette exons (∆PSI ≤ −25) upon Sud C1 treatment. Outliers were discarded, boxes indicate interquartile range (IQR), whiskers extend to 1.5 times IQR. BP features and SF1 binding motif were analyzed in the 3′ 150 nucleotides of each intron. Black lines indicate median values. For some features (SF1 score, SVM BP score and Py-tract score), the difference between the values corresponding to the two introns flanking the alternative exon (upstream–downstream) was calculated, thus comparing the features between the regulated 3′ss and the 3′ss that is used both during exon inclusion and skipping. *; **; ***: Mann–Whitney U-test (two-sided) P value < 0.01, 0.001 and 0.0001, respectively

(Fig. 6a, left), SSA induced retention of nearly threefold higher number of introns than Sudemycins (Fig. 6a, right). Similar effects were observed in total RNAs (Supplementary Fig. 8). Consistently, analysis of the magnitude of splicing changes indicated that while Sudemycins caused slightly stronger effects on exon skipping (Fig. 6b, left), SSA was significantly more effective than Sudemycins in inducing higher levels of intron retention (Fig. 6b, right). Once again, similar tendencies were observed in total RNA (Supplementary Fig. 8). Second, while the three drugs shared a significant number of targets, they also displayed numerous differential effects (Fig. 6c, d). For example, while 70–80% of the intron retention events induced by

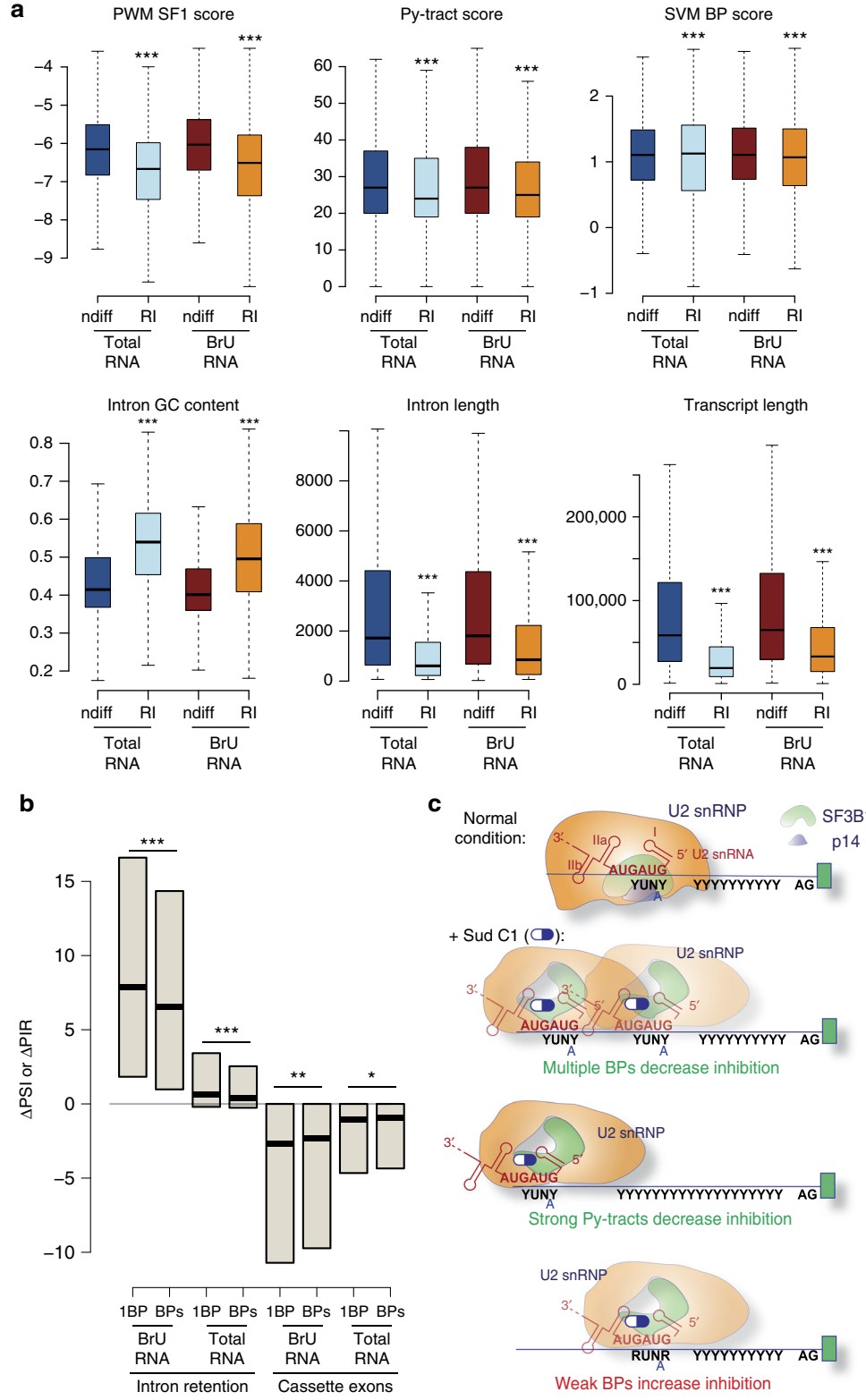

Sudemycins were also induced by SSA, only 52–59% of the exon skipping events were common between these two drug classes. A heatmap of the changes in inclusion levels shows multiple examples of distinct AS changes induced by the three drugs, both in total and in BrU RNAs (Fig. 6d and Supplementary Fig. 8). Of relevance, as shown for the examples in Fig. 5d, e, while SSA showed stronger effects than Sudemycins in a number of cases, the opposite was also observed for other events. RT-PCR assays validated several splicing changes predicted to be common or distinct, both for intron retention and for cassette exon skipping (Fig. 6e).

Finally, sequence features analyses showed that retained introns induced by any of the three drugs are generally shorter, more GC-rich, and harbor weaker BPs compared to non-retained introns (Supplementary Fig. 9). Similar trends are also observed for BPs associated with exon skipping events (Supplementary Figs. 10 and 11). However, these tendencies are less marked for SSA-regulated events, consistent with the more prevalent and extensive intron retention observed upon treatment with this drug. Indeed, minigene analyses showed similar effects of the sequence motifs identified in Fig. 1 upon treatment with the different drugs, but at the same time documented transcript-specific differences, including stronger effects of SSA than Sudemycins on *PDCD10* transcripts (Supplementary Fig. 12). SSA may simply display higher affinity for SF3B1 (leading to more disruptive effects of its function) or induce a different conformation in the protein that is differentially sensitive to the influence of neighboring sequences.

## Discussion

Previous biochemical work[18,46] and recent cryo-EM structures[5,7,8,47,48] revealed that SF3B1 plays a role in the recognition of the BP. Pre-mRNA sequences 3′ of the BP pass through a SF3B1 channel lined up with positively charged amino acids, and pulling of the RNA by the RNA helicase Prp2 at the exit side of the channel induces a collapse of SF3B1 HEAT repeats superhelix domain, bringing the BP adenosine in close proximity to the 5′ end of the intron and thus activating the first catalytic step of the splicing reaction[7,8]. It may be expected that interference with such a fundamental activity would lead to global splicing inhibition and, consequently, dramatic effects on cell viability. However SF3B1 is not only frequently mutated in cancer, with mutations clustering in certain HEAT domain repeats, but the protein is also the molecular target of anti-tumor drugs that bind to the same HEAT domains cavity that embraces the branch adenosine in activated spliceosomes[7,48], arguing for very selective molecular and biological effects.

While previous studies reported a general increase in intron retention upon treatment with SF3B-targeting drugs[49–52], our results suggest that at low drug concentrations splicing inhibition

by these drugs does not cause generalized retention of every newly transcribed intron, but rather leads to differential effects in ss selection, accompanied by AS changes[18,53]. It is conceivable that moderate levels of the drug reduce, but do not completely block, the function of the protein, leading to variable levels of retention of different introns or AS selection.

These selective effects are associated, at least in part, to intrinsic processing properties dictated by the sequence of the pre-mRNA, with discrete and transferable sequence elements working autonomously to modulate drug responses such that even single nucleotide differences can have dramatic effects on drug sensitivity. Drug resistance correlates with the presence of potential functional BP sequences located 5′ of the bona fide BP. Together with our previous observation that SSA induces U2 snRNA base pairing 5′ of the BP[18], these results suggest that drug resistance is associated with the use of functional BPs upon drug-induced relocation of U2 snRNP binding (Fig. 1d). Consistent with this hypothesis, introduction of a consensus BP 5′ of the BP sequence confers drug resistance, but deletion of the BP adenosine from this sequence eliminates drug resistance, as expected if U2 snRNP relocates to a decoy (non-productive)-binding site. Thus, functional BPs 5′ of the normal BP would allow 3′ss activation despite the drug-mediated disabling of the BP proofreading activity of SF3B1 (Fig. 4c).

When multiple BPs are available, the most 3′ BP is used as long as it is not too close to the Py-tract[46]. This may be determined by the interaction between SF3B1 and Py-tract-bound U2AF65[46]. This interaction may be weakened by drug binding to SF3B1, such that U2 snRNA can engage with sites of potential base pairing located farther away from the Py-tract. In this context, it is interesting that mutations in SF3B1 are associated with the use of 5′ BPs and splicing to upstream cryptic 3′ss[13,14], although we find no significant overlap between splicing events activated by SF3B1 mutation and drug-mediated SFB1 inhibition (Fig. 3, Supplementary Fig. 6).

Another sequence determinant associated with drug sensitivity is the extent of base pairing between the BP sequence and U2 snRNA, affecting both intron retention and exon skipping and detected in both steady-state and recently transcribed RNAs. Thus, the high drug sensitivity of *MCL1* intron 1 was abolished when a region containing multiple rather weak BP sequences was replaced by a consensus (yeast-like) BP. Because mutations leading to drug resistance occur within the same internal surface of the HEAT domain repeats that embraces the region of BP/U2 snRNA base pairing[7,8], it is conceivable that drug binding interferes with the accommodation of the base paired RNAs within this cavity[20]. Interference between contacts accommodating drug binding and the U2 snRNA/BP interaction within SF3B1 could be particularly acute for BP regions with limited base pairing with U2 snRNA, or for regions harboring multiple ambiguous BPs (as in human *MCL1*), explaining why BP strength

**Fig. 4** Analysis of transcript features associated with Sud C1-induced intron retention. **a** Sequence features with significant changes between Sud C1-induced retained introns and non-differentially spliced introns. Boxplots represent the distribution of feature values (Methods) in non-differentially spliced introns (ndiff, |ΔPSI| ≤ 5) or retained introns (ΔPIR ≥ 25) upon Sud C1 treatment, for total or BrU-pulsed RNAs. Black lines indicate median values. Outliers were discarded, boxes indicate IQR, whiskers extend to 1.5 times IQR. *; **; ***: Mann–Whitney *U*-test (two-sided) *P* value < 0.01, 0.001 and 0.0001, respectively. **b** Weaker drug effects in the presence of additional BP sequences. Changes in exon inclusion (ΔPSI) or intron retention (ΔPIR) upon Sud C1 treatment are represented for total and BrU-pulsed RNAs, for 3′ss containing a single (1 BP) or multiple (BPs) matches of the BP consensus YNYYRAY within a window of 100 nucleotides from the 3′ss. Exon skipping corresponds to negative ΔPSI values, while intron retention to positive ΔPIR differences. Black lines indicate median values. *; **; ***: Mann–Whitney *U*-test (two-sided) *P* value < 0.01, 0.001 and 0.0001, respectively. **c** Working model summarizing sequence features that enhance or reduce the inhibitory effects of Sud C1 on 3′ss recognition. BP (YUNAY), Py-tract (Y), and AG dinucleotide are indicated in the pre-mRNA, while the BP recognition region (GUAGUA) is indicated in U2 snRNA. SF3B1 and p14 protein components of U2 snRNP are positioned close to the BP/U2 snRNA base paired region but are proposed to adopt an alternative conformation in the presence of the drug

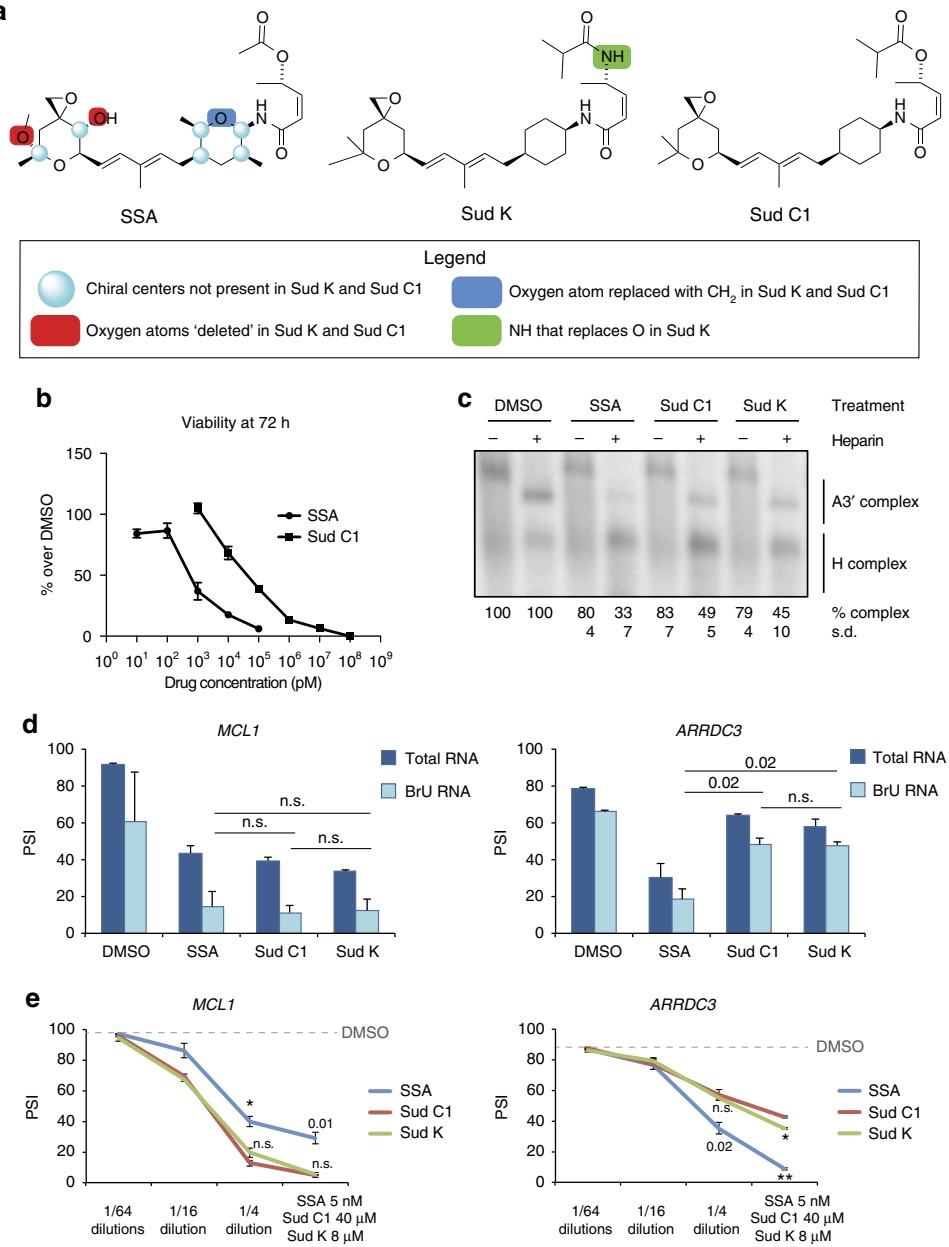

**Fig. 5** Comparison of the activities of sudemycins and SSA. **a** Chemical structures of SSA, Sud C1, and Sud K. Structural differences are highlighted as indicated. **b** Differences in cell toxicity between SSA and Sud C1: viability of Hela cells exposed for 72 h to the indicated drug concentrations was compared to the corresponding control cells treated with DMSO. Error bars indicate s.d. among a minimum of triplicated treatments. **c** Effects of SSA, Sud C1 and Sud K on spliceosome assembly. AdML RNA 3'ss region (40 nucleotides of intron 1 and exon 2) was incubated with HeLa nuclear extracts and treated or not with 5 µg/µl heparin, in the presence of DMSO, 50 nM SSA, 5 µM Sud C1, or 250 nM Sud K. H complex indicates heterogeneous ribonucleoprotein complexes. A3' complex indicates binding of U2 snRNP. **d** Distinguishable effects of SSA, Sud C1, and Sud K on *MCL1* and *ARRDC3* exon skipping. RT-PCR analysis of total and recently transcribed (BrU) RNAs from cells treated for 3 h with DMSO, SSA (5 nM), Sud C1 (10 µM), or Sud K (2 µM) were carried out using primers complementary to exons flanking alternative exons of the *MCL1* and *ARRDC3* genes. **e** Analyses similar to **d** were carried out with serial dilutions of indicated drug concentrations of SSA, Sud C1, and Sud K. Two-sided *t* test *P* values by duplicated treatments are indicated (*; **; ***P value < 0.01, 0.001 and 0.0001, respectively; n.s.: non-significant). Error bars correspond to s.d.

correlates with drug resistance. Comparison between human and mouse *MCL1* shows that a stronger Py-tract can also contribute to resistance to the drug, which may be due to tighter binding of U2AF65 and tighter recruitment of U2 snRNP through the aforementioned interaction between U2AF65 and SF3B1[46]. Previous results from knock down of SF3B complex member PHF5A (also know ad SF3B7 or SF3b14b) also showed that intron

retention is associated with shorter introns harboring more C-rich (weaker) Py-tracts[25].

It seems likely that other factors involved in U2 snRNP recruitment can also modulate the effects of these compounds. For example, the RNA-dependent DEAH-box helicase Prp5 facilitates conformational transitions in U2 snRNA important for U2 snRNP recruitment[54]. These transitions are compromised by

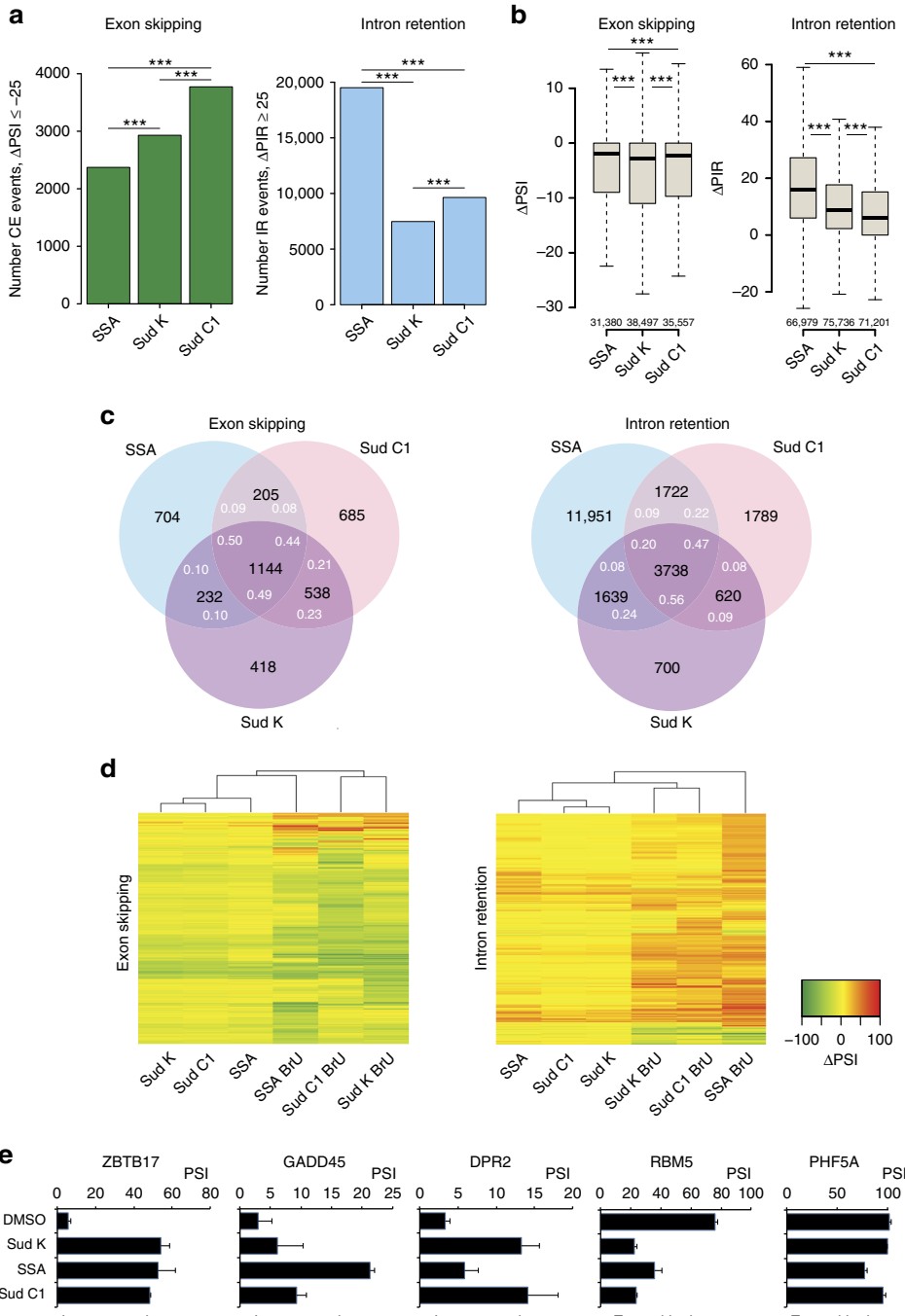

**Fig. 6** Distinct profiles of AS changes induced by SSA and Sud C1 or Sud K. **a** Number of cassette exons and retained introns affected by each of the three drugs in BrU RNA. *; **; ***: $\chi^2$-test $P$ values < 0.01, 0.001, and 0.0001 by comparing regulated events to all detected cassette exons (39,851) and retained introns (76,961), respectively. **b** Distribution of AS changes for cassette exons and retained introns induced by each of the three drugs in BrU RNA. *; **; ***: Mann–Whitney $U$-test (two-sided) $P$ value < 0.01, 0.001, and 0.0001, respectively. Outliers were discarded, boxes indicate interquartile range (IQR), whiskers extend to 1.5 times IQR. Black lines indicate median values. **c** Overlaps between sets of cassette exons and retained introns affected by each of the three drugs, detected in BrU RNA data. Numbers in white indicate the percentage of events in common between the closest drug and the drug(s) in the corresponding overlap section. **d** Clustering analysis of changes in cassette exon skipping or intron retention. Heatmaps represent ΔPSI values of cassette exons upon parallel treatment with the drugs (5 nM SSA, 10 μM Sud C1, and 2 μM Sud K) for 3 h as detected in total or BrU-pulsed RNAs by RNA-seq using VAST-TOOLS (Methods). Only cassette exons or introns with |ΔPSI| or |ΔPIR| ≥ 25 were considered. **e** Examples of validation of differences in AS changes between drugs, predicted by the RNA-seq analyses in **d**. The names of the genes, type of AS, and PSI values under the different conditions are indicated. Treatments were performed in duplicates (5 nM SSA, 10 μM Sud C1, and 2 μM Sud K). Error bars correspond to s.d.

the SF3B1 inhibitor pladienolide[19] and Prp5 activity was also shown to be relevant for the effect of mutations in SF3B1[55,56].

Functions of SF3B1 outside of U2 snRNP complexes, including SF3B reservoirs, have been reported and could also contribute to the transcriptomic changes induced by the drugs. Copy-number variation analysis in cancer show that partial loss of SF3B1 increases the vulnerability of cancer cells to SF3B1 suppression[57], suggesting that a reservoir of SF3B1 exists that can modulate the effects of SF3B1 inhibition. Additional functions of SF3B1 include direct RNA binding to regulatory elements important for AS of particular pre-mRNAs[58] as well as direct association with nucleosomes in chromatin[43]. The latter is particularly interesting because work by Ast and colleagues revealed preferential association of SF3B1 on nucleosomes positioned on GC-rich exons[43], one of the sequence features that we found enriched in exons skipped upon treatment with SF3B1 inhibitory drugs. Thus, the drugs could inhibit SF3B1 association with chromatin and this lead to defective exon recognition, in line with the role of nucleosome positioning on exon definition[39–41]. Also relevant, effects of SF3B-targeting drugs on H3K36 methylation have been reported[53,59], which could contribute to transcriptome changes. Indeed, dynamic acetylation of an H3K4me3 nucleosome positioned on exon 2 of *MCL1* is linked to alternative skipping of this SF3B1-regulated exon[60,61]. SF3B complex components have been also found associated with the epigenetic remodeling complexes Polycomb and SAGA[62–64], adding yet another layer of regulatory functions that can be affected by SF3B1 inhibitors.

SF3B pharmacological inhibition leads to defects in RNA polymerase II Ser2 phosphorylation, 3′ end processing and transcripts release from the site of transcription[51,65,66], suggesting that that the drugs can also directly influence transcription rates and the coupling between transcription kinetics and AS[67].

Moreover, SSA has been shown to induce leakage of pre-mRNAs from the nucleus to the cytoplasm, particularly for transcripts harboring weak BPs and short transcripts length[34], two features that were also captured by our analysis of splicing changes. Long pre-mRNAs, which are more likely to be spliced while still bound to chromatin, may be especially resistant to SSA-induced leakage of pre-mRNAs to the cytoplasm. Drug treatment has been also associated with enlargement of speckles (considered splicing factor storage sites) and pre-mRNA leakage to the cytoplasm[17,68,69], perhaps due to overload of unspliced RNAs in drug-treated cells. Finally, SF3B1 inhibition may affect other categories of transcripts, including some snoRNAs whose processing is linked to splicing and SF3B complex activity[70], as well as U12 introns, given that SF3B1 is also part of the U12 snRNP.

So far, a relatively small number of splicing modulators have been reported[9]. Compounds targeting the SF3B complex have been considered a homogeneous group of drugs sharing a common pharmacophore and molecular target[9,71–73]. In contrast, examples of differential intron retention[74] and our own results indicate that even structurally related drugs can induce a distinct spectrum of splicing changes. Indeed, recent studies identified compounds with therapeutic potential for the treatment of spinal muscular atrophy that selectively influence *SMN2* AS and perhaps a small number of other targets[75,76]. One of these drugs appears to enhance U1 snRNA base pairing with a limited subset of 5′ss in a sequence-dependent manner[76].

One possible explanation for our observations is that variations in drug structure differentially affect the set of interactions within the SF3B1 cavity that embraces the region of U2 snRNA/ BP base pairing[5,7,8,47]. Drugs with different structures may be more or less disruptive for specific 3′ss depending on the base pairing potential of the BP region with U2 snRNA or other sequence features in the region, leading to differential effects of the drugs on 3′ss recognition. We observe that SSA tends to have deeper and wider effects on intron retention than Sudemycins, while the converse appears to apply to exon skipping. It is conceivable that the tighter binding of SSA to SFB31, as illustrated by the lower concentrations required to exert its biological and biochemical effects, imposes stronger restrictions in the fit of the base pairing interactions within the cavity of SF3B1 HEAT domains, and that this results in more general effects. Less tight binding of Sudemycins may be more compatible with a higher variety of interactions leading to 3′ss recognition, leading to wider effects on exon skipping. Competition studies indicate that inactive drugs can still bind to SF3B1, but fail to modulate its activity, indicating that binding to the protein is only part of the molecular events triggered by the drug[71]. This type of pharmacology is reminiscent of agonism vs. antagonism that is observed in many receptors and also suggests that the description of SF3B1 targeted agents as "splicing inhibitors" is an oversimplification of their function[77].

It is also possible that the differential effects of structurally similar drugs are the consequence of a complex interplay between direct and indirect effects. Their similar effects on cancer cell proliferation, however, suggest common key targets. Nevertheless, the observation that structurally related drugs can have versatile effects on the accumulation of mRNA isoforms opens the possibility that small molecules with selective effects on a subset of introns/AS events can be discovered and eventually rationally designed. Given the biological and pathological impact of AS, such modulators could have wide applications to explore gene function and generate novel therapeutic tools.

## Methods

**Drug treatments.** Spliceostatin A was kindly provided by Dr. M. Jurica (UCSC) and Sudemycin K by Drs. K. Makowski and M. Álvarez (Universidad de Barcelona). Drugs were dissolved in DMSO and added in the culture medium or in reaction mixes at the concentrations and times indicated for each experiment.

**Cell culture and gene silencing.** HeLa (ATCC CCL-2) and NIH-3T3 (ATCC CRL-1658) cells were cultured in Glutamax Dulbecco's modified Eagle's medium supplemented with 10% fetal bovine serum and 500 u ml$^{-1}$ penicillin, 0.5 mg ml$^{-1}$ streptomycin. Cells were maintained at 37° under 5% $CO_2$. Short interfering RNAs of interest (stealth siRNA 5′-GACAGCAGAUUUGCUGGAUACGUGA-3′ from Invitrogen for SF3B1 depletion; mission siRNA SASI_Hs01_00162657 5′-GUG UUAAGAGAAGCACUAA[dT][dT]-3′ from Sigma-Aldrich for MCL1 depletion) and Lipofectamine RNAiMAX (Life Technologies) following the manufacturer's recommendations.

**Minigene transfection assays.** Genomic sequences of interest were cloned under a Cytomegalovirus Promoter in a pCMV56 expression vector (Clontech) using KpnI and NotI restriction sites. Mutations were introduced by Gibson cloning. A complete list of minigenes used in this study can be found in Supplementary Table 1. Assays were performed using HeLa cells seeded in 48-well plates (30,000 cells per well) or 96-well plates (10,000 cells per well) and transfected with 3 or 1.5 ng of minigene per well and 0.2 or 0.1 Lipofectamine 2000 per well in 200 or 100 µl Opti-MEM per well, respectively. Hundred nanograms per well of pBluescript plasmid were co-transfected with the minigenes to increase transfection efficiency. NIH-3T3 cells were transfected in 24-well plates (100,000 cells per well) with 25 ng of minigene, 500 ng of pBluescript, and 0.5 µl of Lipofectamine 2000 per well. Drug treatments were performed after overnight transfection in 200, 100, or 50 µl of total medium for 24-, 48-, and 96-wells plates, respectively.

**RNA extraction and RT-PCR.** poly(A) mRNA was extracted using oligo-dT-coated 96-well plates (mRNA catcher PLUS, Life Technologies). One third of purified poly(A) mRNAs was reverse-transcribed in a 40 µl volume with 0.25 µl of Superscript III (Life Technologies). Reverse transcription was carried out in the presence of 2.5 µM oligo-dT (Sigma-Aldrich) and 250 ng of random primers (Life Technologies). PCR reactions were carried out using GoTaq DNA polymerase (Promega).

For amplification of minigene-specific transcripts, oligonucleotide primers complementary to transcribed regions within vector sequences were used (PT1 and PT2, Supplementary Table 2). Amplification of endogenous transcripts was achieved using primers complementary to regions of the mRNA not included in the minigene (e.g., constitutive exons adjacent to the sequence included in the minigene).

PCR products were resolved by capillary electrophoresis using the Labchip GX Caliper workstation (Caliper, Perkin Elmer) and HT DNA 5K LabChip chip (Perkin Elmer).

**Spliceosome A3′ complex formation**. This assay was carried out as previously described[18]: each splicing reaction contained 1 µl of RNA mix (premix for 10 reactions: 100 fmol RNA corresponding to the 3′ end of AdML intron 1 and the downstream exon, 4 µl creatine phosphate 0.5 M, 1 µl ATP 100 mM, 1 µl MgCl$_2$ 270 mM, up to 10 µl with buffer D), 3 µl of HeLa nuclear extracts (Cilbiotech), 1 µl of drug or DMSO, 2 µl of 15% polyvynil alcohol, prewarmed at 30 °C, and buffer D 0.1 M KCl up to 9 µl, to obtain standard splicing conditions (3 mM MgCl$_2$, 1.1 mM ATP, 22 mM creatine phosphate, 1.67% polyvinyl alcohol). ATP and creatine phosphate were replaced with buffer D 0.1 M KCl for the -ATP control.

Buffer D contains 20 mM HEPES-KOH pH 7.9; 0.2 mM EDTA, 20% glycerol, 1 mM DTT, 0.01% NP40, complemented with 0.1 M KCl and 1 mM DTT (freshly added).

The reactions were set up in a 48-wells microplates and incubated at 30 °C for 15′. Subsequently, 5 mg ml$^{-1}$ heparin and 2.2 µl of 6× DNA loading dye (20 mM Tris-HCl at pH 7.5, 0.25% bromophenol blue, 0.25% xylene cyanol, 30% glycerol) were added and the reactions were incubated for 10′ at room temperature. The products were subsequently loaded on a 1.5% low-melting agarose (Invitrogen) gel in 50 mM Tris base and 50 mM glycine buffer for 90′ at 4 °C and run at 75 V. Gels were fixed in 10% methanol and 10% acetic acid for 10′ at room temperature, dried for at least 3 h at 50 °C and exposed overnight to a phosphorimager screen. The intensity of the band corresponding to complex A3′ over the signal of the whole well was measured by ImageJ and normalized to the control condition (after subtracting the background signal from each measurement).

**BrU-IP and RNA-Seq analysis**. HeLa cells (800,000 in a 6 cm plate) were co-treated with 2 mM BrU (5-Bromouridine, Sigma-Aldrich) and 10 µM Sud C1, 2 µM Sud K, or 5 nM SSA A for 3 h. RNA was extracted using the Maxwell RNAeasy kit (Promega). Total RNA was kept as input and for standard RNA sequencing. Ten micrograms of total RNA were immunoprecipitated with a mouse anti-BrU antibody (B2531, Sigma), as previously described[78] except that no tRNA was added: 2 µl of antibody were incubated with 20 µl Protein G Dynabeads (Invitrogen) per sample for 1 h at 4 °C in 1 ml RSB-100 buffer (10 mM Tris-HCl, pH 7.4, 100 mM NaCl, 2.5 mM MgCl$_2$ and 0.4% (v/v) Triton X-100). After three washes of the beads with 1 ml of RSB-100 buffer, beads were resuspended in 150 µl RSB-100 with 40 U RNasin (Promega). RNA was incubated for 1 min at 80 °C, added to the beads and incubated for 1 h at 4 °C. After three washes with 1 ml of RSB-100 buffer, the RNA bound to the beads was eluted by direct addition of 300 µl RLT buffer (Qiagen RNeasy mini kit). The mix was then heated at 80 °C for 10′ and after centrifugation the BrU-labeled RNA present in the supernatant was purified using the RNeasy mini kit (Qiagen).

RNA-Seq was performed on a Illumina HiSeq 2000 at the CRG Genomics Unit. Stranded mRNA-seq libraries (TruSeq) were prepared and sequenced under conditions of 2 × 125 bp paired-end reads. Duplicates were sequenced for each condition. For BrU-IP RNA, no poly(A) selection was performed before library preparation.

**Bioinformatic analyses**. Splicing analyses of RNA-Seq data were performed using VAST-TOOLS (Vertebrate Alternative Splicing and Transcription Tools): https://github.com/vastgroup/vast-tools[79] (Supplementary Data 1 and 2) and analyzed downstream using custom scripts invoking SVM-BPfinder[37] for scoring BP and Py-tracts. In addtition, the BP strength was approximated by the maximal binding score of a PWM for the human SF1[36] within the 3′ 150 intronic nucleotides. GO analyses were performed using GOrilla (GO enRIchment anaLysis and visuaLizAtion tool). Heatmaps were obtained using heatmap.2 function from R package gplots.

**Cytotoxicity assays**. A total of 1000 or 2500 HeLa cells per well were seeded in 96-well transparent plates the day before drug treatment or other experimental conditions. 20, 44 and 68 h after treatment, the cell medium was replaced with medium containing 10 µM Resazurin and cells incubated at 37 °C for 4 h. Fluorescence was measured with an Infinite 200 PRO series multi-plate reader (TECAN) set to 530 and 590 nm as fluorescence excitation and emission wavelengths, respectively.

**Branch point mapping from total RNA**. Complementary DNA (cDNA) was synthetized using 500 ng–1 µg of total cellular RNA in the presence of random hexamers and Superscript III reverse transcriptase. PCR amplification was performed using a forward primer annealing in the 3′ region of the intron (but 5′ of potential BPs) and a reverse primer complementary to the 5′ end of the intron, in order to obtain products only if amplification occurs across the 2′-5′ lariat structure (primers are listed in Supplementary Table 2). Thirty cycles of PCR were performed with 2 µl of cDNA and 15 s of elongation time. Products were diluted 1:500, re-amplified with nested primers (following the same PCR conditions) and loaded on 6% polyacrylamide gels. Bands of potential interest were excised from the gel, eluted by adding 20 µl of water, and shaking at 50 °C for 30 min. After brief centrifugation, supernatants were used for Sanger sequencing reactions.

**Data availability**. RNA-seq data have been deposited in the ArrayExpress database under accession number E-MTAB-6060. The data that support the findings of this study are available from the corresponding author upon request.

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

## Acknowledgements

Spliceostatin A was kindly provided by M. Jurica and Sudemycin K by K. Makowski and M. Álvarez. Work in J.V. laboratory is supported by Fundación Botín, Banco de Santander through its Santander Universities Global Division, the European Research Council, AGAUR, Spanish Ministry of Economy and Competitiveness and the Centre of Excellence Severo Ochoa. Work in M.I. laboratory is supported by European Research Council, Spanish Ministry of Economy and Competitiveness and the Centre of Excellence Severo Ochoa. We also thank Panagiotis Papasaikas, Pierre de la Grange, Monica Nafria Fedi, Álvaro Moreno, Anna Ribó Rubio, Elisabeth Daguenet for technical help and Eduardo Eyras, Mayka Sánchez, Josep Vilardell and members of our research groups for constructive feedback and discussion.

## Author contributions

The study was conceived by J.V. and designed by L.V. and J.V. Experiments were performed by L.V. and bioinformatics analyses by A.G. and M.I.; T.W. provided Sudemycin C1 and feedback. The manuscript was written and edited by: L.V., A.G., M.I., and J.V. All co-authors reviewed and approved the submission.

**Additional information**

**Competing interests:** The authors declare no competing financial interests.

