## [Peer Review File · Nature Communications]

Reviewers' comments:

Reviewer #1 (Remarks to the Author):

Vigevani et al analyze the molecular basis of different anti tumor drugs (spliceostatin and sudemycin).

The manuscript has two novel and significant findings: a) it determines sequence motifs around the branchpoint as well as multiple sequence features as contributors for sudemycin effects and b) shows differences between the chemically highly related compounds.

1) the first part of the paper is a highly focused analysis of MCL1 splicing, whereas the second part is a genome wide analysis using Sudemycin C1, K and SSA. Key findings in Fig 1B and 1C should be repeated using sudeK and SSA to allow for better comparisons, especially in the element mapping fig. 1B. A minor point is that the boxes in Fig 1B (E3, E2, E1) look like exon symbols, and will be clearer if another shape (oval) is used. I assume that the exon in Fig 1B is #7, which should be indicated.

2) The box plot annotation (fig 3, 4) should be explained in the figure legends. Assuming standard annotation was used (whiskers are max, min and the box is the IQR), differences in Fig 3C SF1, SVM, Py tract, and GC content do not look significant as indicated by the stars, which needs to be clarified.

3) In Fig. 5, for the MCL1 and ARRDC3 genes, and the assembly assays (5C, D) all three compounds should be used to allow for a systematic comparison.

4) The various mechanistically details in the discussion should be made into a schematic figure, allowing to visualize the different features. Also, the question why are sudemycins selective for cancer cells should be touched in the discussion, as sudeK and C1 show similar effects on cancer cells, but have different effects on splicing.

5) the result of the RNAseq data should be added in a supplement or database, ideally top splicing changes as xls files.

Minor points:

The writing needs more attention, references perriman and folco on p 17 are not formatted;

branchpoint should be uniform, right now it is BP, branchpoint, branchsite, sudemycin C1 or sude C1

The differences in chemical structures described in the text could be highlighted in Figure 5A

At the 5'ss, usually GU is considered completely conserved, not GU/C

Reviewer #2 (Remarks to the Author):

The manuscript by Vigevani et al, is well written and reports on studies elucidating the molecular mechanisms of action of a set of drugs targeting the SF3B component of the spliceosome. These drugs are clinically relevant since tumor cells with aberrant splicing activity, either by mutations in the SF3B gene or misregulated expression of splicing factors, are more sensitive to these drugs. The experiments performed are cleverly designed and well executed and contributes greatly to the mechanistic understanding of these drugs and of the splicing process itself.

Specific Comments

- 1) Are the drugs directly reversible or do cells have to synthesize new SF3B proteins in order to regain splicing function?
- 2) It was suggested that “the function of SF3B is inhibited to different extents dependent on the drug concentration”. It could also be that SF3B molecules are fully inhibited by the drug but the reservoir of SF3B protein is large enough so that at lower doses a pool of SF3B proteins still operates on the stronger splice junctions. Alternatively, only some cells are taking up the inhibitor at lower doses but many take it up at high doses.
- 3) The labeling of “nascent RNA” to study the “kinetics” of splicing is clever but the choice of one labeling time point (3h) makes the data difficult to interpret. After a 3 hour labeling time it is expected that “older” RNA (labeled in the beginning of the 3-hour period) would be fully spliced while younger RNA would be in the process of splicing. Also, there would be plenty of time for quality control mechanisms (RNA exosome) to clean up splicing mistakes and eliminate such RNAs. These possibilities should be addressed and the rationale for using a 3-hour BrU labeling period stated.
- 4) It would be interesting to see the overlap between the introns from total RNA and BrU RNA upon drug treatment (Fig 4 A).
- 5) Is it possible that BrU incorporation into the RNA causes splicing alterations? Can this be ruled out?

Reviewer #3 (Remarks to the Author):

This study from Vigevani examines the determinants of splicing sensitivity to a class of anti-tumor drugs that includes Sudemycins and Spliceostatin A. These drugs target the key spliceosomal component SF3B1. SF3B1 is itself mutated in a variety of tumors and pre-cancerous states and is the subject of intense study. Both drug treatment and SF3B1 mutation have both been shown to alter the splicing of only certain introns, which is somewhat surprising as the protein is essential for splicing. This has led to a model that particular mis-spliced targets drive the tumor phenotype. Other studies have indicated that tumor cells might be particularly sensitive to the drugs because of their specific effects rather than a general splicing inhibition. The authors now examine the differential sensitivity of introns to the drugs. Through a series of well-designed chimeric reporter genes they show that specific sequence elements surrounding the branch point confer sensitivity or resistance to the drug. Notably, these elements resemble additional functional or non-functional branch points. Using RNAseq to profile splicing across the transcriptome in drug treated cells they comprehensively identify drug sensitive splicing events and identify several features that skew differently in the drug sensitive targets, most notably sequences in the branch point and 3' splice site. Interestingly they find that despite their common target protein, the two drugs sudamycin and spliceostatin inhibit splicing of quite different sets of transcripts. Finally, they show that U2-containing spliceosomal A complexes, which have SF3B1 bound to the RNA, are more sensitive to heparin induced disassembly in the presence of drug. This sensitivity to heparin correlates with whether the splice site is sensitive to the drug in vivo.

This is a well-executed study that makes an important contribution to our understanding of the mechanism of SF3B1 targeted drugs. Although the conclusions from the genomewide analysis are at this point limited, they provide a useful starting point for further experiments, and will find wide readership. I have only minor suggestions.

1. Some of the text was hard to follow and I was often confused by whether a sequence being described was a true functional branchpoint or a non-functional one, which I guess was being described as a decoy. For example in Figure 1B, there is a branchpoint marked in sequence E1 and in the gene diagram above. It was not clear to me upon first reading that they were different.
2. As they discuss, in SF3B1 mutants, a pattern of incorrect splicing to upstream branchpoints has been described by others. Do they see a similar shift with drug treatment? They may mention this but in all the discussion of degenerate branchpoints, I missed it.
3. On page 11 (Figures 3 and 4) different properties are described for drug sensitive introns (weaker branchpoints and py-tracts etc) but the meaning of these findings are hard to discern. While the averages of these properties change, they still exhibit a broad distribution within both the sensitive and insensitive populations. One suspects that these differences are a consequence of something else that has not yet been measured. Can they take this analysis further to more

closely identify the important features? Have they measured motif frequencies in the key regions they have identified upstream and downstream from the branchpoint. Given their mutagenesis data, they might find differences in branchpoint-like motifs that more clearly predict drug sensitivity.

4. The biochemical analysis of drug sensitive splice sites also doesn't get taken very far. If they test some of the mutants developed in Figure 1 for heparin sensitivity, can they show a correlation between strength of binding in heparin and drug resistance?

5. Editing issues: There are misformatted references on page 17 (Perriman and Folco). On page 19, do they mean "agonism versus antagonism"?

Point by Point Rebuttal to the Reviewers' comments

Reviewer #1

Vigevani et al analyze the molecular basis of different anti tumor drugs (spliceostatin and sudemycin).

The manuscript has two novel and significant findings: a) it determines sequence motifs around the branchpoint as well as multiple sequence features as contributors for sudemycin effects and b) shows differences between the chemically highly related compounds.

We appreciate the reviewer's praise of our data as novel and significant and specially his/her constructive and useful feedback.

1) the first part of the paper is a highly focused analysis of MCL1 splicing, whereas the second part is a genome wide analysis using Sudemycin C1, K and SSA. Key findings in Fig 1B and 1C should be repeated using sudeK and SSA to allow for better comparisons, especially in the element mapping fig. 1B. A minor point is that the boxes in Fig 1B (E3, E2, E1) look like exon symbols, and will be clearer if another shape (oval) is used. I assume that the exon in Fig 1B is #7, which should be indicated.

As requested by the reviewer, we have reproduced key findings of Figure 1 using the three drugs (SSA, Sud C1 and Sud K) and show these results in Supplementary Figure 12. Two conclusions can be derived from these new data:

a) the drug-protective effect of the E1/E2/E3 sequence elements, as well as the effects of consensus branch points, decoys and strong polypyrimidine tract are similar for the three drugs, suggesting similar general mechanisms of action.

b) nevertheless, while SSA and Sudemycins have similar effects on MCL1 minigene transcripts, SSA is clearly more active on transcripts of PDCD10 minigenes under the same experimental conditions. These results nicely complement the data of Figures 5 and 6, providing additional evidence for the differential effects of these drugs on different RNAs despite their similar general mechanism of action.

These new results and conclusions are now described and discussed (p. 12).

Figure 1B: following the reviewer's advice, boxes representing elements E3, E2 and E1 have been replaced by ovals (to avoid confusion with exons) and exon numbers are now also indicated.

2) The box plot annotation (fig 3, 4) should be explained in the figure legends. Assuming standard annotation was used (whiskers are max, min and the box is the IQR), differences in Fig 3C SF1, SVM, Py tract, and GC content do not look significant as indicated by the stars, which needs to be clarified.

The reviewer is correct in his/her interpretation of the boxplot annotation, which we now explain in Figure legends 3 and 4, as suggested. We have applied the Mann-Whitney-U test for our comparisons and report the statistical significance of the results using the star code. While the boxes and whiskers often overlap, the large number (hundreds to thousands) of data points allow to report statistically robust, even if small, differences between the distributions.

3) In Fig. 5, for the MCL1 and ARRDC3 genes, and the assembly assays (5C, D) all three compounds should be used to allow for a systematic comparison.

New Figure 5C shows similar effects of the three compounds on spliceosome assembly assays, recapitulating previous published results from our lab using SSA (Corrionero et al, 2011) and extending them to both Sudemycin C1 and K. The results indicate that the drugs have little effect on complex A formation in the absence of heparin, while they reduce complex formation in the presence of heparin, arguing that the three drugs act by destabilizing U2 snRNP assembly. The three drugs were also included in the experiments of panels 5D and E. The results indicate, as expected, that Sudemycin C1 and K display very similar effects on MCL1 and ARRDC3 transcripts, while SSA displays stronger effects than Sudemycins on ARRDC3 transcripts, but not on MCL1 transcripts, further arguing for the distinct effects of SSA vs Sudemycins on alternative splicing. The new data are described and discussed in the text (p. 11).

4) The various mechanistically details in the discussion should be made into a schematic figure, allowing to visualize the different features. Also, the question why are sudemycins selective for cancer cells should be touched in the discussion, as sudeK and C1 show similar effects on cancer cells, but have different effects on splicing.

New Figure 4C provides a schematic summary of the different features found to influence drug responses, which are discussed in the text (pp 13-14). We now also discuss various possible explanations for the selective effects of Sudemycins in cancer cells and for the similar anti-cancer effects of drugs that display distinguishable effects on splicing (pp. 18).

5) the result of the RNAseq data should be added in a supplement or database, ideally top splicing changes as xls files.

We have uploaded the results of the RNA-seq experiments in ArrayExpress (Accession E-MTAB-6060) and in addition now provide compressed txt files (Supplementary Data 1 and 2) containing the output of *vast-tools* analysis for splicing (Data 1) and gene expression (Data 2) under the different experimental conditions.

Minor points:

The writing needs more attention, references Perriman and Folco on p 17 are not formatted; branchpoint should be uniform, right now it is BP, branchpoint, branchsite, sudemycin C1 or sude C1

We apologize for these mistakes, which have been corrected in the revised manuscript. We have also adopted a uniform nomenclature for branch point (BP) and drugs.

The differences in chemical structures described in the text could be highlighted in Figure 5A

New Figure 5A highlights the chemical / chiral groups that structurally distinguish between the three drugs used in the study.

At the 5'ss, usually GU is considered completely conserved, not GU/C

This sentence has been deleted to accommodate the text to the length limits.

Reviewer #2

The manuscript by Vigevani et al, is well written and reports on studies elucidating the molecular mechanisms of action of a set of drugs targeting the SF3B component of the spliceosome. These drugs are clinically relevant since tumor cells with aberrant splicing activity, either by mutations in the SF3B gene or misregulated expression of splicing factors, are more sensitive to these drugs. The experiments performed are cleverly designed and well executed and contributes greatly to the mechanistic understanding of these drugs and of the splicing process itself.

We sincerely thank the reviewer for his/her positive opinion of our work and the excellent constructive feedback.

Specific Comments

1) Are the drugs directly reversible or do cells have to synthesize new SF3B proteins in order to regain splicing function?

This is an interesting point that we have not directly addressed in our manuscript because previous evidence indicates that SF3B complex components can be washed away from the beads containing biotinylated SSA, arguing that the interaction is reversible (Kaida et al., Nature Chem. Biol. 2007). This was an interesting observation, since the highly reactive epoxide group present as part of the common pharmacophore of SF3B inhibitors could potentially form a covalent bond with the pharmacological target, as discussed by Kaida and colleagues. In addition, other groups have observed that cultured cells lose alternative splicing

changes induced by the drug after changing the culture medium (Koga et al., Plos ONE 2014; Convertini et al., Nuclei Acids Research 2014).

This does not mean, however, that other processes and cellular decisions induced by the drugs are reversible. For example, chromatin modifications can persist long after drug removal (Convertini et al., Nuclei Acids Research 2014) and of course commitment to apoptosis can be also irreversible (Kashyap et al., 2015). Our own results (presented in Supplementary Figure 1) show that inhibition of cell proliferation persists long after 3h or 24h of treatment with Sud C1 at concentrations of 2 or 20 μM .

Collectively, these data indicate that while the immediate effects of the drugs on SF3B1 on splicing may be reversible, phenotypic changes may persist, for example leading to cell proliferation arrest and/or apoptosis.

2) It was suggested that “the function of SF3B is inhibited to different extents dependent on the drug concentration”. It could also be that SF3B molecules are fully inhibited by the drug but the reservoir of SF3B protein is large enough so that at lower doses a pool of SF3B proteins still operates on the stronger splice junctions. Alternatively, only some cells are taking up the inhibitor at lower doses but many take it up at high doses.

This is also an intriguing question and interesting alternative explanations. While addressing them experimentally is beyond the scope of this report, we now mention this among the alternative scenarios for drug action (p. 16).

3) The labeling of “nascent RNA” to study the “kinetics” of splicing is clever but the choice of one labeling time point (3h) makes the data difficult to interpret. After a 3 hour labeling time it is expected that “older” RNA (labeled in the beginning of the 3-hour period) would be fully spliced while younger RNA would be in the process of splicing. Also, there would be plenty of time for quality control mechanisms (RNA exosome) to clean up splicing mistakes and eliminate such RNAs. These possibilities should be addressed and the rationale for using a 3-hour BrU labeling period stated.

The rationale for using a 3 hour labeling time is that effects on splicing start to be detectable in total RNA at this time point and therefore we considered that they should be obvious in recently transcribed RNA as well, thus ensuring the usefulness of the material obtained in these rather expensive experiments. While it is true that a BrU pulse of 3 hours contains a mixture of unprocessed, fully processed and partially processed transcripts, there is substantial enrichment in not fully processed RNAs that can be used to compare with steady-state transcripts, which are essentially fully processed. As described in more detail in the response to the next point, we do observe a much larger number of intron retention events in BrU-labeled compared to total RNAs, while the overlap between the two classes of RNAs is very high when comparing changes in inclusion of alternative cassette exons (which necessarily correspond to fully processed transcripts). In other words, while a shorter pulse could have provided

a more straightforward distinction between just transcribed and "older" RNA -as the referee points out-, the 3h pulse provides useful information about recently transcribed RNAs as well as a reference for comparing fully processed transcripts. We now explain this point in the text (pp. 8-9).

4) It would be interesting to see the overlap between the introns from total RNA and BrU RNA upon drug treatment (Fig 4 A).

New Supplementary Figure 7 shows the overlap between the effects of drug treatments on total and BrU-pulsed RNAs. The results indicate that

1) while more intron retention events are detected in BrU-labeled RNA than in total RNA (as expected if BrU allows to detect a higher proportion of unprocessed transcripts), most of the events detected in total RNA are also detected in BrU-labeled RNA: 84% (SSA), 74% (Sud C1) and 70% (Sud K) which is quite reasonable considering that this level of overlap is often found in biological replicas, and that p values from Chi-square tests indicate highly significant overlaps ($< 2.2 \times 10^{-16}$).

2) when looking at processed transcripts, e.g. when looking at increased exon skipping events, a category which is highly sensitive to drugs, specially for Sudemycins, the overlap is even higher: 88% (SSA), 90% (Sud C1) and 85% (Sud K).

Our conclusion is that BrU allows to detect, as expected, a larger number of splicing events but that the majority of events detected in total RNA are also detected in BrU-pulsed RNA. Therefore BrU does not induce a substantially new set of splicing changes, a conclusion that helps to address also the next point raised by the reviewer (point 5).

5) Is it possible that BrU incorporation into the RNA causes splicing alterations?

The results discussed above to some extent address this point: if BrU would induce an entirely different set of splicing changes, then the overlaps between changes induced by the drug in total and BrU-pulsed RNAs would not be expected to be so high. But of course it could be argued that BrU induces a **new** set of **additional** splicing changes and this is the reason why the BrU-mediated changes are in larger numbers.

To address this possibility, we compared the profiles of alternative splicing for intron retention and exon skipping events between BrU-labeled and total RNA in control (DMSO-treated) conditions. For exon skipping events we observed that

- 75 exons were more skipped in Total RNA than BrU-labeled RNA
- 25.109 exons were not differentially spliced between the two types of RNA samples ($-5 < \Delta\text{PSI} < 5$)
- 19 exons were more included in Total RNA than in BrU-labeled RNA

These results show that 99.6% of the cassette exons were not differentially spliced in BrU-labeled samples compared to total RNA samples, indicating that BrU does not significantly alter the patterns of RNA processing.

Not surprisingly, we found more differences for intron retention:

- 1 intron was more retained in total RNA than in BrU-labeled RNA
 - 7033 introns were not differentially retained between the two types of RNAs
 - 11705 introns were more retained in BrU-labeled RNA compared to total RNA
- This is expected considering that BrU-labeled RNAs are particularly enriched in unspliced and partially processed, recently transcribed RNAs, while such species are a minority in total RNA, as it is composed mainly of fully processed RNAs in steady-state.

Thus, the strong coincidence in cassette exon splicing profiles (necessarily reporting processed RNAs) between BrU-pulsed and total RNAs reveals that BrU does not by itself induce significant splicing alterations.

We now refer to the lack of effects of BrU on processed RNAs in the text (p. 9).

Reviewer #3

This study from Vigevani examines the determinants of splicing sensitivity to a class of anti-tumor drugs that includes Sudemycins and Spliceostatin A. These drugs target the key spliceosomal component SF3B1. SF3B1 is itself mutated in a variety of tumors and pre-cancerous states and is the subject of intense study. Both drug treatment and SF3B1 mutation have both been shown to alter the splicing of only certain introns, which is somewhat surprising as the protein is essential for splicing. This has led to a model that particular mis-spliced targets drive the tumor phenotype. Other studies have indicated that tumor cells might be particularly sensitive to the drugs because of their specific effects rather than a general splicing inhibition. The authors now examine the differential sensitivity of introns to the drugs. Through a series of well-designed chimeric reporter genes they show that specific sequence elements surrounding the branch point confer sensitivity or resistance to the drug. Notably, these elements resemble additional functional or non-functional branch points. Using RNAseq to profile splicing across the transcriptome in drug treated cells they comprehensively identify drug sensitive splicing events and identify several features that skew differently in the drug sensitive targets, most notably sequences in the branch point and 3' splice site. Interestingly they find that despite their common target protein, the two drugs sudamycin and spliceostatin inhibit splicing of quite different sets of transcripts. Finally, they show that U2-containing spliceosomal A complexes, which have SF3B1 bound to the RNA, are more sensitive to heparin induced disassembly in the presence of drug. This sensitivity to heparin correlates with whether the splice site is sensitive to the drug in vivo.

This is a well-executed study that makes an important contribution to our understanding of the mechanism of SF3B1 targeted drugs. Although the

conclusions from the genomewide analysis are at this point limited, they provide a useful starting point for further experiments, and will find wide readership. I have only minor suggestions.

We very much thank the referee for her/his positive feedback and suggestions.

1. Some of the text was hard to follow and I was often confused by whether a sequence being described was a true functional branchpoint or a non-functional one, which I guess was being described as a decoy. For example in Figure 1B, there is a branchpoint marked in sequence E1 and in the gene diagram above. It was not clear to me upon first reading that they were different.

We apologize for the lack of clarity. We have done our best to streamline the text and clarify the terms utilized and we hope that the concepts are more clear now. Regarding Figure 1B, we understand the concern of the reviewer and have tried to clarify it by labeling differently the “BP” (i.e. the functional BP in normal conditions) and the “potential BP” (i.e. a potentially functional alternative BP).

2. As they discuss, in SF3B1 mutants, a pattern of incorrect splicing to upstream branchpoints has been described by others. Do they see a similar shift with drug treatment? They may mention this but in all the discussion of degenerate branchpoints, I missed it.

This is an important point, which we have addressed as follows:

a) our RNA-Seq analysis revealed that regulated alt 3' ss are a minority of events (Figure 3A and Supplementary Figure 6) and we found minimal overlap between these cases and the ones previously reported by Alsafadi et al., Nature Communications 2016 and Darman et al., Cell Reports 2015 associated with SF3B1 mutations. Consistent with this, a recent publication from Teng et al., Nature Communications 2017 reported that using the same pipeline that detected cryptic 3' ss activated by SF3B1 mutations in Darman et al., 2015 detected, in contrast, mainly exon skipping and intron retention upon drug treatment.

b) we have carried out RT-PCR assays designed to detect cryptic 3' splice site activation in genes expressed in our cells and that were shown to display cryptic 3' splice site branch point activation upon SF3B1 mutation, specifically in the genes DIP2A and ZDHHC16, as reported by Darman et al., Cell Reports 2015. We could not detect cryptic 3' splice site activation in these transcripts upon drug treatment.

We conclude that drug treatments do not generally lead to the activation of upstream 3' splice sites observed upon expression of SF3B1 mutants. This may not be unexpected given the distinct location of the mutations and of the drug binding pocket in the structure of SF3B1 (e.g. Teng et al Nature Communications 2017). We now mention these results in the text (p. 14).

3. On page 11 (Figures 3 and 4) different properties are described for drug

sensitive introns (weaker branchpoints and py-tracts etc) but the meaning of these findings are hard to discern. While the averages of these properties change, they still exhibit a broad distribution within both the sensitive and insensitive populations. One suspects that these differences are a consequence of something else that has not yet been measured. Can they take this analysis further to more closely identify the important features? Have they measured motif frequencies in the key regions they have identified upstream and downstream from the branchpoint. Given their mutagenesis data, they might find differences in branchpoint-like motifs that more clearly predict drug sensitivity.

Again this is a very relevant point. We followed the reviewer's suggestion and searched for sequence motifs and other features, including GC content as well as branch point and SF1 motifs around (+- 75 nt) predicted BP positions. We found clear differences in GC content between affected and non-affected introns. After stratification, i.e. sub-sampling for matching the GC content of the intron sets to be compared, we generated motif RNA maps and found a considerable enrichment of the BP motif YNYYRAY in non-affected introns over introns affected by Sud C1 and Sud K. This finding strengthens our hypothesis that introns with more BPs tend to be less affected by the tested drugs. We show these data in Supplementary Figure 5 and discuss them in the text (p. 10). At this point, however, we believe that the effect of the drugs may be best explained by a combination of various sequence features. We are developing approaches to combine multiple features in a predictive model, but this is work still in progress that requires substantial efforts beyond this first report.

4. The biochemical analysis of drug sensitive splice sites also doesn't get taken very far. If they test some of the mutants developed in Figure 1 for heparin sensitivity, can they show a correlation between strength of binding in heparin and drug resistance?

The problem that we have faced when trying to address this issue is that for some reason MCL-1 transcripts do not form spliceosome complexes in vitro under conditions in which other pre-mRNAs (e.g. AdML, Fas intron 5) do. To some extent this is also the case for PDCD10 transcripts, although for this RNA some U2 snRNA-dependent complexes are detectable. However the effects of addition of the drugs have proved to be very variable. For example, at one point we observed (see Figure below) formation of two U2 snRNA-dependent complexes in the presence of E1E2E3 elements, and one of these complexes appeared to be destabilized in the presence of Sudemycin C1, while the other was not.

These results would be in agreement with the idea that two types of U2 snRNP assemblies can occur on wild type PDCD10 (but not in the absence of E1E2E3), one of which being drug-sensitive while the other one is not, matching our proposal that the E1/E2/E3 elements contain functional branch points and provide resistance to the drug. However this result turned out to be not reproducible every time we do the experiment and therefore, after numerous attempts to find out the origin of this variability, we regretfully had to abandon this line of experiments. We also used psoralen crosslinking and found that drug treatment led to reproducible enhancement of U2 snRNA crosslinking in PDCD10 transcripts but not in RNAs lacking the E1/E2/E3 element (data not shown), consistent with relocation of U2 snRNP to these sequence elements. However it was not possible to isolate enough amount of psoralen adducts to map the sites of crosslinking by primer extension and therefore the observation could not be followed up either.

In our experience carrying out biochemical experiments in nuclear extracts with alternatively spliced pre-mRNAs is often complicated by the relatively weak splicing signals characteristic of regulated spliced regions and it is sometimes just not feasible -at least currently- to explore mechanisms using these technologies.

5. Editing issues: There are misformatted references on page 17 (Perriman and Folco). On page 19, do they mean “agonism versus antagonism”?

We apologize for these mistakes, which have been corrected in the revised version. Thank you.

Reviewers' Comments:

Reviewer #1 (Remarks to the Author):

na, all questions have been addressed

Reviewer #2 (Remarks to the Author):

The authors have adequately addressed all my previous comments.

Reviewer #3 (Remarks to the Author):

The authors have carefully addressed all of the concerns raised in the previous review. The paper is an interesting and substantial contribution to our understanding of drugs targeting SF3B1 and is ready for publication.

Response to referees

We want to thank the three referees for their positive opinion on our revised manuscript. The three referees accepted the manuscript without further comments.